# Unlocking Clinical Potential: Beyond Single-to-Tri-Phase CT with Dynamic Fusion for Liver Tumor Segmentation

## Abstract

Liver tumor segmentation is essential for treatment planning and disease monitoring. Most existing methods rely on single-phase computed tomography (CT), they often suffer from low contrast and incomplete lesion depiction. Contrast-Enhanced CT (CECT) offers multiple imaging phases: arterial (ART), portal venous (PV), and delayed (DL), which provide complementary anatomical and functional information. This study begins with a systematic quantitative evaluation of each enhanced phase using standard segmentation models to investigate their individual contributions and validate phase-specific clinical insights. Guided by this analysis, a Multi-phase Attention Deep Fusion Network (`MADF-Net`) is proposed to hierarchically integrate ART, PV, and DL features across the input, feature, and decision levels. Experiments on the clinically collected multi-phase liver lesion (MPLL) dataset (the largest and most clinically comprehensive multi-phase liver cancer CECT dataset) demonstrate that the proposed method achieves state-of-the-art segmentation performance. `MADF-Net` achieves a Dice score of $\mathbf{78.65\%}$, which is $\mathbf{9.39\%}$ higher than the best single-stage baseline, by deeply fusing information from three phases, and consistently improves across all evaluation metrics. Our codes are available at https://anonymous.4open.science/r/ICLR26_unlocking_clinical_potential-EFE8/.

## 1 Introduction

Liver tumor segmentation is a critical task in quantitative medical image analysis, providing essential morphological and spatial information for surgical planning, radiotherapy, and post-treatment monitoring (Bilic et al., 2023). With the advent of deep learning, fully convolutional networks

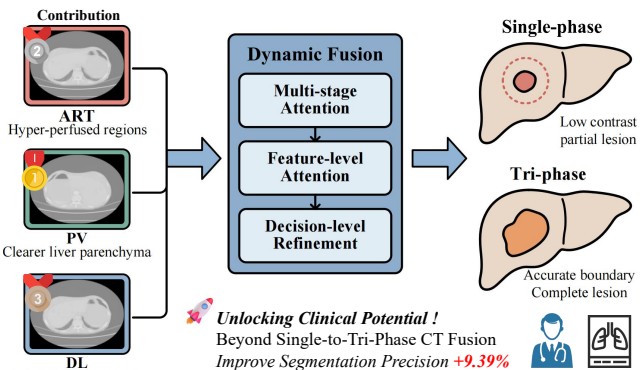

Figure 1: Motivation of our multi-phase CT fusion strategy. Different CT phases provide complementary information: ART highlights vessels, PV improves lesion–parenchyma contrast, and DL captures delayed enhancement. Their integration yields more accurate, robust liver tumor segmentation.

(FCNs), particularly U-Net and its variants (Ronneberger et al., 2015; Ren & Li, 2025; Du et al., 2022; Huang et al., 2017), have achieved notable success in automated segmentation tasks. These models extract features at either the 2D slice level or the 3D volumetric level, enabling robust representation learning for complex anatomical structures (Li et al., 2018).

However, the majority of existing methods rely solely on single-phase Computed Tomography (CT) images (Bilic et al., 2023; Wang et al., 2023; Hatamizadeh et al., 2022; Cao et al., 2022), often ignoring the phase-specific characteristics inherent in clinical imaging protocols (Jun et al., 2023). Due to low tissue contrast and resolution limitations, single-phase methods frequently fail to achieve the precision required for clinical deployment

(Song et al., 2023; Zhang et al., 2024). Contrast-Enhanced CT (CECT), which captures dynamic changes in tissue attenuation following contrast agent administration, provides a valuable alternative by acquiring images at multiple time points—typically including the non-contrast (NC), arterial (ART), portal venous (PV), and delayed (DL) phases (Chi et al., 2013).

Among these, the NC phase offers a baseline anatomical information, but lacks enhancement patterns relevant to the tumor vasculature and lesion contrast, and is therefore generally not emphasized in liver tumor segmentation studies (Ni et al., 2024; Xu et al., 2021; Liu et al., 2024). The ART phase captures early vascular features, highlighting hyper-perfused regions and enhancing lesion boundary delineation (Kulkarni et al., 2021; Urban et al., 2000). The PV phase provides clearer liver parenchyma and structural completeness, facilitating more accurate segmentation (Kulkarni et al., 2021; Schneider et al., 2014). The DL phase captures delayed enhancement and washout phenomena, aiding in the identification of fibrotic or hypo-perfused tumors (Monzawa et al., 2007; Lim et al., 2002). The complementary nature of these phases offers a compelling opportunity for improved segmentation through multi-phase fusion (As shown in Figure 1). Therefore, how to effectively extract and fuse the features from different phases has attracted the attention of many researchers.

Existing multi-phase fusion strategies can be broadly categorized into three types (Zhang et al., 2021b): (1) *Input-level fusion* (Ouhmich et al., 2019), where multiple phases are concatenated as input and processed via a shared encoder; (2) *Feature-level fusion* (Zhang et al., 2021b; Zhu et al., 2022; Zhang et al., 2023; Hazirbas et al., 2016; Liu et al., 2023), which extracts features from each phase independently before combining them at intermediate layers; and (3) *Decision-level fusion* (Sun et al., 2017; Raju et al., 2020), where each phase is processed by a separate network and results are fused at the output level. While these approaches have shown potential, they often suffer from limitations such as insufficient modeling of nonlinear inter-phase relationships (Sun et al., 2017; Zhang et al., 2023), reduced reliability in ambiguous or low-contrast regions, and vulnerability to missing-phase scenarios common in clinical workflows (Xu et al., 2021; Zhu et al., 2022). Moreover, many existing methods treat all phases with equal importance during fusion, overlooking their distinct clinical value and the complementary information they offer (Xu et al., 2021; Zhong et al., 2024; Qiao et al., 2024). This results in suboptimal performance, especially in cases with blurred lesion boundaries or small lesions. **Therefore**, *how to effectively fuse multi-phase CT features while leveraging their individual strengths and mitigating their limitations remains an open challenge (Jiang et al., 2020).*

In this paper, we begin by systematically evaluating the segmentation performance of each enhanced CT phase using standard deep learning models. Our quantitative analysis reveals that the PV phase contributes most significantly to segmentation accuracy, consistent with its known clinical role. Guided by this observation, we propose a novel framework, Multi-phase Attention Deep Fusion Network (`MADF-Net`), to exploit the complementary advantages of the ART, PV, and DL phases through hierarchical fusion. `MADF-Net` introduces full-stage attention-based fusion across the input, feature, and decision levels, enabling deep inter-phase information interaction. Extensive experiments on the MPLL dataset demonstrate that our method achieves state-of-the-art segmentation performance, reaching a Dice score of $78.65\%$ when using all three phases, representing a $9.39\%$ improvement over the best single-phase baseline, confirming its robustness and generalizability.

Our contributions are as follows:

❶ We conduct a comprehensive quantitative analysis of liver tumor segmentation across different CT phases and demonstrate the predominant contribution of the PV phase, providing both empirical and clinical insights.

❷ We propose `MADF-Net`, a multi-phase attention-based fusion network that integrates ART, PV, and DL phase features at multiple stages, enhancing liver tumor segmentation performance through deep inter-phase feature interaction.

❸ Extensive experiments on a newly collected multi-phase liver lesion (MPLL) benchmark (the largest and most clinically comprehensive multi-phase liver cancer CECT dataset) demonstrate that the proposed method achieves state-of-the-art liver tumor segmentation performance.

## 2 RELATED WORKS

**Single-Phase Based Liver Tumor Segmentation.** Deep learning has significantly advanced single-phase liver tumor segmentation in CT images. Ronneberger et al. (Ronneberger et al., 2015)

introduced U-Net, whose encoder-decoder structure with skip connections became foundational, effectively capturing both local details and global context for handling low contrast and fuzzy boundaries. H-DenseUNet (Li et al., 2018) enhanced feature reuse through hybrid dense connections and achieved state-of-the-art results on the LiTS2017 dataset. Variants such as UNet++ (Zhou et al., 2018) further improved efficiency and multi-scale accuracy, particularly for small tumor detection. However, CNN-based models often struggled with global context in complex tumor structures. To address this, TransUNet (Chen et al., 2021) combined CNNs for low-level feature extraction with Transformers for global dependency modeling. UNETR (Hatamizadeh et al., 2022) and UNETR++ (Shaker et al., 2024) integrated global context and local detail via a Transformer-based U-shaped architecture, achieving strong performance on 3D CT tasks. Nevertheless, single-phase methods still suffer from limited tissue contrast and resolution, resulting in information loss and reduced clinical applicability.

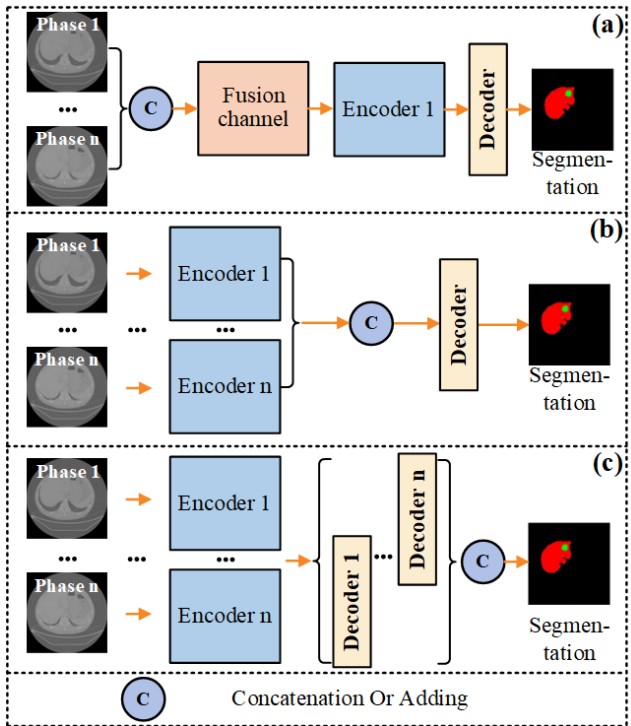

Figure 2: Multi-phase fusion method of enhanced CT. (a), (b), and (c) correspond to input-level, feature-level, and decision-level fusion architectures, respectively.

**Multi-Phase Based Liver Tumor Segmentation.** Recently, an increasing number of studies have investigated how to leverage multi-phase CT information to improve liver tumor segmentation performance. Multiphase fusion is typically performed at one of three stages: input-level, feature-level, or decision-level fusion (Zhang et al., 2021b), referred to as single-stage fusion in this paper. Alternatively, fusion can occur across multiple stages, which we define as multi-stage fusion. ❶ **Single-stage Fusion.** An early example of single-stage input-level fusion was proposed by Ouhmich et al. (Ouhmich et al., 2019), who concatenated PV and ART phase images as input to a U-Net, significantly improving tumor segmentation performance over single-phase training. Feature-level fusion is currently the most active research area. Zhou et al. (Zhou et al., 2019) introduced a dual-path 3D fully convolutional network with cross-phase skip connections to enable dense information exchange. Wu et al. (Wu et al., 2019) treated non-contrast and enhanced CT scans equally and applied feature-level fusion at selected U-Net layers. In decision-level fusion, features are independently extracted from each phase and fused at higher layers. Raju et al. (Raju et al., 2020) proposed an integrated joint and semi-supervised training strategy that leveraged limited plain and enhanced CT data to achieve robust cross-domain segmentation. Despite progress in single-stage fusion, these methods still face challenges such as information loss and limited ability to capture phase-specific characteristics. ❷ **Multi-stage Fusion.** Feature-level and decision-level combinations currently dominate multi-stage fusion network designs (Ni et al., 2024; Liu et al., 2024; Zhu et al., 2022; Kuang et al., 2024). PA-ResSeg (Xu et al., 2021) introduced intra- and inter-phase attention mechanisms to capture both channel-wise dependencies and cross-phase interactions, embedding attentional modules at each encoder layer to fuse multi-scale information from ART and PV phases. Building on this, SA-Net (Zhang et al., 2021b) added a spatial aggregation module for encoding-stage interaction and an uncertainty correction module at the decision stage to refine fuzzy tumor boundaries. To address spatial misalignment in multi-phase CT, Zhang et al. (Zhang et al., 2023) incorporated differentiable deformation operations (Jaderberg et al., 2015) for enhanced feature alignment. Raju et al. (Raju et al., 2020) proposed a joint and semi-supervised training strategy that effectively leveraged limited non-contrast and enhanced CT data, though at the cost of increased training time. HRadNet (Liang et al., 2023) utilized a feature

pyramid and a metadata fusion layer to incorporate clinical features such as tumor size and patient age, improving generalizability. However, most existing multi-stage approaches adopt only two fusion stages and still suffer from potential information loss, limiting segmentation accuracy and robustness. To address this, we propose a three-stage fusion network designed to better preserve information throughout the extraction and fusion process.

# 3 PRELIMINARY

To fully exploit the complementary anatomical and pathological information provided by the three phases of CECT, we propose a novel three-stage fusion framework, named Multi-phase Attention Deep Fusion Network (`MADF-Net`). This section first introduces preliminary knowledge on fusion strategies, and then describes the proposed `MADF-Net`. As shown in Figure 2, three common fusion strategies are illustrated. A detailed pseudocode description of the overall procedure is provided in Appendix A.

**Input-level Fusion.** As shown in Figure 2 (a), this strategy concatenates images from different phases (Phase 1, Phase 2, ..., Phase $n$) along the channel dimension at the input stage to form a unified input tensor. To enhance flexibility, we introduce learnable phase-wise modulation weights $\{\alpha_i\}_{i=1}^n$ and a normalization operator $\mathcal{N}(\cdot)$:

$$\boldsymbol{I}_{\text{input}} = \mathcal{N}\left(\Big\|_{i=1}^n \big(\alpha_i \cdot \Gamma(\boldsymbol{I}_i) + \beta_i \cdot \mathbf{1}_{H \times W \times C}\big)\right), \alpha_i = \frac{\exp(\theta_i)}{\sum_{j=1}^n \exp(\theta_j)}, \tag{1}$$

where $\Gamma(\cdot)$ denotes intensity standardization, $\boldsymbol{I}_i \in \mathbb{R}^{H \times W \times C}$ is the $i$-th phase image, $\theta_i$ are learnable logits and $\|$ denotes channel-wise concatenation. This formulation adaptively highlights more informative phases while suppressing noisy ones.

**Feature-level Fusion.** As shown in Figure 2 (b), this strategy integrates features from multiple phases by using attention-guided gating and nonlinear projections. Let $\boldsymbol{E}_i \in \mathbb{R}^{H' \times W' \times d}$ be the feature map extracted from the $i$-th phase. We compute phase attention maps $\boldsymbol{A}_i$ from global descriptors $\boldsymbol{g}_i$ via a softmax-normalized MLP, and then fuse features as:

$$\boldsymbol{E}_{\text{fusion}} = \Psi\left(\sum_{i=1}^n \Big(\underbrace{\frac{\exp(\mathbf{W}_a \boldsymbol{g}_i + \mathbf{b}_a)}{\sum_{j=1}^n \exp(\mathbf{W}_a \boldsymbol{g}_j + \mathbf{b}_a)}}_{\boldsymbol{A}_i} \odot \underbrace{(\mathbf{W}_v * \boldsymbol{E}_i + \mathbf{b}_v)}_{\tilde{\boldsymbol{E}}_i}\Big)\right), \boldsymbol{g}_i = \text{GAP}(\boldsymbol{E}_i), \tag{2}$$

where $*$ denotes convolution, $\text{GAP}(\cdot)$ is global average pooling, and $\Psi(\cdot)$ is a residual refinement block. This design allows adaptive semantic fusion guided by global context cues.

**Decision-level Fusion.** As shown in Figure 2 (c), this strategy constructs separate segmentation heads for each phase, and aggregates the resulting predictions $\{\boldsymbol{S}_i\}_{i=1}^n$ based on their confidence. We employ an uncertainty-aware soft weighting scheme with temperature scaling:

$$\boldsymbol{S}_{\text{final}} = \sum_{i=1}^n \Big(\underbrace{\frac{\exp\big(-\tau \cdot \mathcal{H}(\boldsymbol{S}_i)\big)}{\sum_{j=1}^n \exp\big(-\tau \cdot \mathcal{H}(\boldsymbol{S}_j)\big)}}_{w_i}\Big) \cdot \underbrace{\sigma(\boldsymbol{S}_i)}_{\text{probability map}}, \mathcal{H}(\boldsymbol{S}_i) = -\frac{1}{|\Omega|}\sum_{p \in \Omega}\sum_c \boldsymbol{S}_i^{(p,c)} \log \boldsymbol{S}_i^{(p,c)},$$
$$\tag{3}$$

where $\mathcal{H}(\cdot)$ computes the spatial entropy over pixel set $\Omega$, and $\tau$ controls weight sharpness. This formulation emphasizes confident predictions and suppresses noisy ones for decision-level fusion.

# 4 METHODOLOGY

To address the challenge of liver tumor segmentation using multiphase CT data, we propose a unified framework, `MADF-Net`, that integrates input-level, feature-level, and decision-level fusion. As shown in Figure 3, the network consists of two parallel branches (main and auxiliary) with symmetric encoder-decoder structures, enabling hierarchical feature aggregation across three CT phase.

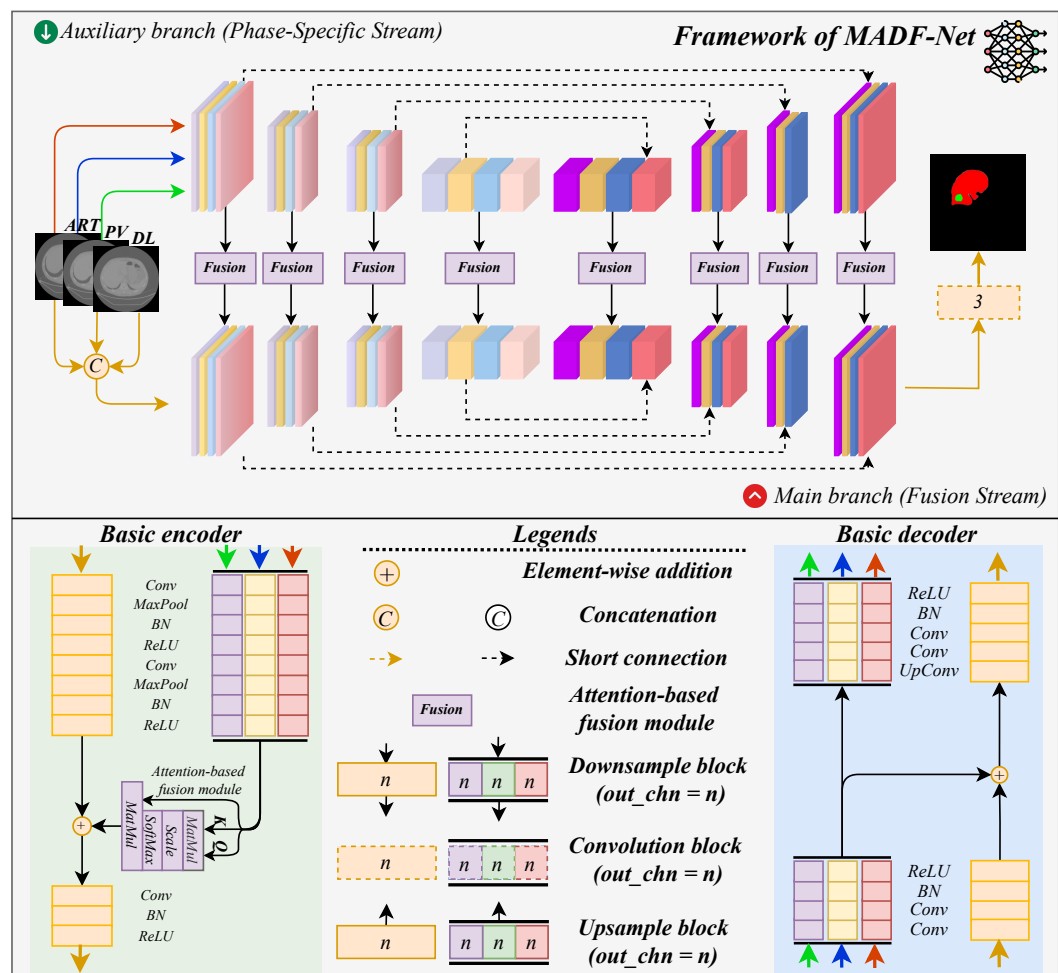

Figure 3: Overview of MADF-Net. Three single-phase CT inputs (ART, PV, DL) are processed via parallel main/auxiliary branches. Fusion occurs at three levels: (1) Input-level (concatenation of phases in main branch); (2) Feature-level (cross-branch fusion in encoder/decoder blocks); (3) Decision-level (final result aggregation).

**Input-Level Fusion: Multi-Phase Data Initialization.** Given the three CT phases $\boldsymbol{I}_{ART}$, $\boldsymbol{I}_{PV}$, and $\boldsymbol{I}_{DL}$, the main branch performs early-stage fusion by channel-wise concatenation to directly expose the encoder to cross-phase correlations: $\boldsymbol{I}_{fusion} = \text{Concat}(\boldsymbol{I}_{ART}, \boldsymbol{I}_{PV}, \boldsymbol{I}_{DL}) \in \mathbb{R}^{H \times W \times 3C}$, where $H$ and $W$ denote the spatial resolution and $C$ denotes the number of channels per phase. In parallel, the auxiliary branch independently forwards each phase through three isolated sub-encoders: $\boldsymbol{I}_{phase}^{(p)} = \{\boldsymbol{I}_{ART}, \boldsymbol{I}_{PV}, \boldsymbol{I}_{DL}\}$, $p \in \{1, 2, 3\}$, preserving phase-specific characteristics that might otherwise be suppressed by early fusion.

**Encoder Blocks: Hierarchical Feature Extraction with Cross-Branch Fusion.** Both branches comprise four encoder blocks indexed by $l \in \{1, \ldots, 4\}$, progressively downsample spatial resolution while expanding feature dimensionality (e.g., $C \to 4C \to 16C \to 64C$). Encoder block performs following operations: **(I) Self-Attention-based Phase Reweighting.** For auxiliary features $\boldsymbol{X}_{ART}^{(l)}$, $\boldsymbol{X}_{PV}^{(l)}$, and $\boldsymbol{X}_{DL}^{(l)}$, a shared self-attention module computes attention weights across phases:

$$\boldsymbol{\alpha}^{(l)} = \text{softmax}\left(\frac{\boldsymbol{Q}^{(l)} \cdot \left(\boldsymbol{K}^{(l)}\right)^{\top}}{\sqrt{d_k}}\right), \quad \boldsymbol{Q}^{(l)} = \boldsymbol{W}_Q \boldsymbol{X}^{(l)}, \ \boldsymbol{K}^{(l)} = \boldsymbol{W}_K \boldsymbol{X}^{(l)}, \quad (4)$$

where $\boldsymbol{W}_Q, \boldsymbol{W}_K \in \mathbb{R}^{d_k \times d_{in}}$ are learnable projections and $\boldsymbol{X}^{(l)} = \left[\boldsymbol{X}_{ART}^{(l)}, \boldsymbol{X}_{PV}^{(l)}, \boldsymbol{X}_{DL}^{(l)}\right]$. The reweighted auxiliary feature is then: $\boldsymbol{X}_{aux}^{(l)} = \sum \alpha_p^{(l)} \odot \boldsymbol{X}_p^{(l)}, p \in \{ART, PV, DL\}$

**(II) Feature-level Cross-Branch Fusion.** After obtaining the reweighted auxiliary representation, we further integrate it with the main branch feature $\boldsymbol{X}_{main}^{(l)}$. To this end, we design a *gated residual summation* mechanism:

$$\boldsymbol{X}_{enc}^{(l)} = \sigma\Big(\boldsymbol{W}_g * [\boldsymbol{X}_{main}^{(l)}, \boldsymbol{X}_{aux}^{(l)}]\Big) \odot \boldsymbol{X}_{main}^{(l)} + \Big(1 - \sigma\Big(\boldsymbol{W}_g * [\boldsymbol{X}_{main}^{(l)}, \boldsymbol{X}_{aux}^{(l)}]\Big)\Big) \odot \boldsymbol{X}_{aux}^{(l)}, \quad (5)$$

where the gating factor is adaptively determined by the concatenated representations from both branches. This allows the model to dynamically balance their contributions, avoiding redundancy and gradient dilution caused by naive concatenation or summation. Specifically, when the gate approaches 1, the model emphasizes high-level semantics from the main branch, while values closer to 0 highlight fine-grained cues from the auxiliary branch.

Nevertheless, simply stacking the above fusion operation across multiple depths may hinder gradient propagation, thus limiting the representation capacity of deeper layers. To address this, we introduce a *residual preservation regularization* to facilitate cross-layer information flow. Concretely, at the $l$-th layer, the fused feature $\boldsymbol{X}_{enc}^{(l)}$ is enhanced with a gated residual connection from previous layer:

$$\hat{\boldsymbol{X}}_{enc}^{(l)} = \boldsymbol{X}_{enc}^{(l)} + \lambda \cdot \Big(\alpha^{(l)} \odot \hat{\boldsymbol{X}}_{enc}^{(l-1)} + \big(1 - \alpha^{(l)}\big) \odot f\Big(\hat{\boldsymbol{X}}_{enc}^{(l-1)}\Big)\Big), \quad l > 1, \quad (6)$$

where $\lambda$ is a learnable global balancing coefficient, and $\alpha^{(l)}$ is a layer-wise gating vector that adaptively controls the trade-off between direct residual propagation and a transformed path. The function $f(\cdot)$ denotes a lightweight non-linear mapping (e.g., a convolutional projection or an MLP). This design ensures that shallow features can effectively penetrate deeper layers to improve gradient flow, while the nonlinear transformation path enriches cross-layer feature diversity.

**Decoder Blocks: Multi-Scale Feature Reconstruction.** The decoder consists of four blocks that mirror the encoder structure, progressively upsampling the fused representations $\boldsymbol{X}_{enc}^{(l)}$ back to the original resolution. Each decoder block not only restores spatial resolution but also selectively incorporates complementary information from shallow layers through gated skip connections. Specifically, at each stage $l$, we compute:

$$\boldsymbol{Y}^{(l-1)} = \text{ReLU}\Big(\boldsymbol{G}^{(l)} \odot \text{UpConv}\big(\boldsymbol{X}_{enc}^{(l)}\big) + \big(1 - \boldsymbol{G}^{(l)}\big) \odot \mathcal{F}_{att}\big(\boldsymbol{X}_{enc}^{(l-1)}, \text{UpConv}(\boldsymbol{X}_{enc}^{(l)})\big)\Big), \quad (7)$$

where $\boldsymbol{G}^{(l)}$ is a learned gating map, $\text{UpConv}(\cdot)$ denotes an upsampling convolution block with batch normalization and activation, and $\mathcal{F}_{att}(\cdot, \cdot)$ is an attention-based fusion module for shallow-deep interaction. This stage-wise design ensures that high-resolution details from earlier layers are progressively blended with the deep semantic context from later layers.

To further enhance multi-scale consistency and stabilize gradient flow, we augment the reconstruction with a residual-preserving multi-scale aggregation term:

$$\hat{\boldsymbol{Y}}^{(l-1)} = \boldsymbol{Y}^{(l-1)} + \mu \cdot \Big(\sum_{k=1}^{K} \beta_k^{(l)} \odot \text{UpConv}^{(k)}\Big(\boldsymbol{X}_{enc}^{(l-k)}\Big)\Big), \quad (8)$$

where $\mu$ is a learnable global scaling factor, $\beta_k^{(l)}$ are adaptive weights normalized by a softmax constraint, and $\text{UpConv}^{(k)}(\cdot)$ denotes $k$-step hierarchical upsampling operators. This formulation explicitly aggregates contextual evidence from multiple encoder depths, enabling the decoder to reconstruct fine details while preserving long-range semantic dependencies.

In summary, the decoder leverages a combination of gated skip fusion, residual-preserving connections, and multi-scale aggregation to ensure both spatial fidelity and semantic consistency. Such a design alleviates the common issue of blurred boundaries in dense prediction tasks, while also enhancing robustness against vanishing gradients during backpropagation.

**Decision-Level Fusion: Final Segmentation Output.** Finally, the outputs of the main and auxiliary decoders are aggregated to produce the segmentation mask. Specifically,

$$\boldsymbol{O}_{final} = \sigma\Big(\boldsymbol{W}_{out} * \big[\boldsymbol{O}_{ART}, \boldsymbol{O}_{PV}, \boldsymbol{O}_{DL}, \boldsymbol{O}_{fusion}\big] + \boldsymbol{b}_{out}\Big), \quad (9)$$

where $\boldsymbol{O}_{ART}, \boldsymbol{O}_{PV}, \boldsymbol{O}_{DL}$ are the three auxiliary outputs, $\boldsymbol{O}_{fusion}$ is the main-branch output, and $\boldsymbol{W}_{out}, \boldsymbol{b}_{out}$ denote the parameters of the final convolutional projection. This decision-level fusion enforces complementary exploitation of both phase-specific and cross-phase knowledge, yielding a precise tumor segmentation mask.

## 5 EXPERIMENTS

**Dataset Curation (Multi-phase Dataset).** The multi-phase liver lesion (MPLL) dataset, consists of $952, 601$ 2D slices with liver disease from the "Anonymous Authoritative Hospitals (Information will be made public after the paper is accepted)". The dataset includes patients aged between 9 and 72 years, and the number of axial slices per scan varying from 48 to 777. This is the largest and most valuable multi-phase CECT liver cancer dataset to date, all the images in MPLL dataset contain three enhanced phases (ART, PV, and DL). The registered images were annotated using ITK-SNAP software by two experienced attending radiologists, and subsequently reviewed by a third attending radiologist to ensure the accuracy and consistency of the annotations. All data have been anonymized and contain only image information. The MPLL dataset has received approval from the institutional ethics committee under certification number 2022-BE(H)-194. Figure 6 shows example images from the datasets. The training, validation, and test splits (7:1:2, following previous work (Jiang et al., 2023)), along with image dimensions and other details, are summarized in Table 1. A more detailed description is provided in Appendix B.

Table 1: **Dataset characteristics.**

| Dataset | Attribute | Value |
|---------|-----------|-------|
| **MPLL** | Phase | ART, PV, DL |
| | Slice thickness | 0.62mm–5.0mm |
| | Slice resolution | $512 \times 512$ |
| | Disease type | ABS, HCC, HEM, ICC, Lipoma |

**Evaluation Metrics.** We employed the Dice Similarity Coefficient (DSC), Jaccard Similarity Coefficient (JSC), Average Symmetric Surface Distance (ASSD), and 95% Hausdorff Distance ($HD_{95}$) (Jiang et al., 2025) to evaluate the experimental results. In the experiments on single-phase (1P), two-phase (2P) and three-phase (3P) input, we additionally employed Volume Overlap Error (VOE) and Relative Volume Difference (RVD) as supplementary evaluation metrics.

**Implementation Details.** All models were trained for 100 epochs with a batch size of 8. The Stochastic Gradient Descent (SGD) optimizer was adopted with a learning rate of 0.01 and 4 parallel data loading workers. Data augmentation techniques include horizontal flipping (with probability 0.5) and vertical flipping (with probability 0.5), and no post-processing is used.

The proposed method was implemented on a Linux 5.4.0 system using PyTorch 1.13.1. All experiments were conducted on two NVIDIA GeForce RTX 3090 GPUs (24 GB × 2), providing sufficient computational resources for efficient model training and evaluation.

### 5.1 MAIN RESULT

Table 2: Quantitative comparison of segmentation performance across different phase combinations (relative to 3P, based on numerical differences).

| Input Phases | DSC(%)↑ | JSC(%)↑ | $HD_{95}$↓ | ASSD↓ |
|--------------|---------|---------|-----------|-------|
| 1P (PV) | $69.26_{\downarrow 9.39}$ | $64.08_{\downarrow 10.73}$ | $40.298_{\uparrow 13.492}$ | $16.958_{\uparrow 6.222}$ |
| 2P (ART+PV) | $76.09_{\downarrow 2.56}$ | $71.41_{\downarrow 3.40}$ | $28.838_{\uparrow 2.032}$ | $15.659_{\uparrow 4.923}$ |
| 3P (ART+PV+DL) | **78.65** | **74.81** | **26.806** | **10.736** |

**Obs. ❶: Phase Combination Analysis.** We conducted three groups of experiments to explore the optimal input combination for 1P, 2P, and 3P settings. As listed in Table 2, the number of input phases increases, segmentation accuracy improves accordingly, demonstrating the effectiveness of the proposed `MADF-Net` in leveraging complementary information across multiple imaging phases. Specifically, the best result of 1P input (Zheng et al., 2024b) using only the PV phase achieves a DSC of 69.26% and a JSC of 64.08%. When the ART phase is added to form the 2P input (ART+PV), both DSC and JSC show moderate improvements to 76.09% and 71.41%, respectively, while the $HD_{95}$ drops significantly from 40.298 to 28.838, indicating better boundary localization. The 3P input (ART+PV+DL) achieves the best overall performance, with the highest DSC (78.65%) and JSC (74.81%), along with the lowest $HD_{95}$ (26.80) and Average ASSD of 10.73. These results suggest that the additional information from the ART and DL phases enhances both global overlap and local boundary accuracy. Compared to the 1P setting, the 3P input yields substantial gains of 9.39% in DSC and 10.73% in JSC, underscoring the benefit of multi-phase fusion in capturing diverse tumor characteristics.

**Obs. ❷: Comparison of Single Phase Performance.** We conducted a segmentation performance comparison using different single-phase input images on the MPLL dataset, evaluating several state-of-the-art models, including FANet (Tomar et al., 2022), GRENet (Wang et al., 2023), ASSNet (Zheng et al., 2024a), TransUNet (Chen et al., 2021), KiU-Net (Valanarasu et al., 2021),

Table 3: Quantitative comparison of different methods on the MPLL dataset. Best results are **bold**, and differences relative to PV are shown as colored arrows.

| Methods | DSC(%)↑ | | | Jaccard(%)↑ | | | HD$_{95}$ (mm)↓ | | | ASSD (mm)↓ | | |
|---------|---------|---------|---------|---------|---------|---------|---------|---------|---------|---------|---------|---------|
| | ART | PV | DL | ART | PV | DL | ART | PV | DL | ART | PV | DL |
| FANet | 67.56$_{↓1.52}$ | **69.08** | 66.01$_{↓3.07}$ | 61.98$_{↓1.11}$ | **63.09** | 60.74$_{↓2.35}$ | 60.384$_{↑17.529}$ | **42.855** | 62.083$_{↑19.228}$ | 30.300$_{↑10.107}$ | **20.193** | 25.206$_{↑5.013}$ |
| GRENet | 66.59$_{↓1.67}$ | **68.26** | 66.21$_{↓2.05}$ | 60.14$_{↓0.94}$ | **61.08** | 59.91$_{↓1.17}$ | 60.651$_{↑3.390}$ | **57.261** | 65.412$_{↑8.151}$ | 28.069$_{↑4.059}$ | **24.010** | 30.982$_{↑6.972}$ |
| ASSNet | 67.91$_{↓1.35}$ | **69.26** | 66.49$_{↓2.77}$ | 60.01$_{↓4.07}$ | **64.08** | 60.07$_{↓4.01}$ | 45.260$_{↑4.962}$ | **40.298** | 59.946$_{↑19.648}$ | 22.074$_{↑5.116}$ | **16.958** | 25.031$_{↑8.073}$ |
| TransUNet | 67.51$_{↓0.97}$ | **68.48** | 67.15$_{↓1.33}$ | 58.65$_{↓2.58}$ | **61.23** | 60.53$_{↓0.70}$ | 55.960$_{↓−0.339}$ | **55.301** | 58.594$_{↑3.293}$ | 26.983$_{↓0.346}$ | **27.329** | 24.972$_{↓2.357}$ |
| KiU-Net | 65.92$_{↓1.70}$ | **67.62** | 65.62$_{↓2.00}$ | 59.64$_{↓1.24}$ | **60.88** | 59.42$_{↓1.46}$ | 62.902$_{↑3.642}$ | **59.260** | 65.956$_{↑6.696}$ | 30.821$_{↑2.178}$ | **28.643** | 27.983$_{↓0.660}$ |
| AttUNet | 66.01$_{↓2.51}$ | **68.52** | 65.85$_{↓2.67}$ | 59.82$_{↓1.84}$ | **61.66** | 59.61$_{↓2.05}$ | 60.913$_{↑5.289}$ | **55.624** | 66.583$_{↑10.959}$ | 31.098$_{↑4.117}$ | **26.981** | 27.973$_{↓−3.008}$ |

and AttUNet (Chen et al., 2023). The results are summarized in Table 3. Across all models, segmentation performance was consistently better on the PV phase compared to the ART and DL phases. For example, with FANet, the DSC and JSC on PV were 1.52% and 1.11% higher than on ART, and 3.07% and 2.35% higher than on DL, respectively. In terms of boundary metrics, the HD$_{95}$ and ASSD on PV were 17.529 and 10.107 lower than on ART, and 19.228 and 5.013 lower than on DL, indicating more accurate boundary localization. These results suggest that segmentation outputs on the PV phase more closely match the ground truth. In clinical contexts, this may be attributed to the PV phase offering more distinct grayscale contrast between tissues, as well as between lesions and normal structures (Ni et al., 2024; Lam et al., 2017; Al-Battal et al., 2024). This contrast enhancement is particularly beneficial in cases with ambiguous or highly heterogeneous tumor boundaries (Liu et al., 2024).

**Obs. ❸: 2P Fusion: ART and PV.** Single-phase experimental results indicate that segmentation performance is high when using the PV and ART phases as inputs. Based on this observation, we conducted 2P (A+P) experiments on the MPLL dataset, comparing the proposed `MADF-Net` with several state-of-the-art multi-phase segmentation methods, including MAML (Zhang et al., 2021a), MW-UNet (Zhu et al., 2022), SA-Net (Zhang et al., 2021b), PA-ResSeg (Xu et al., 2021), and MCDA-Net (Kuang et al., 2024). As listed in Table 4, `MADF-Net` achieves superior performance, improving the DSC metric by 5.27%, 4.71%, 4.18%, 1.90%, and 0.01% over the five comparison methods, respectively. It also consistently outperforms all other methods in terms of HD$_{95}$, demonstrating its ability to exploit complementary information across imaging phases to enhance both segmentation accuracy and boundary localization.

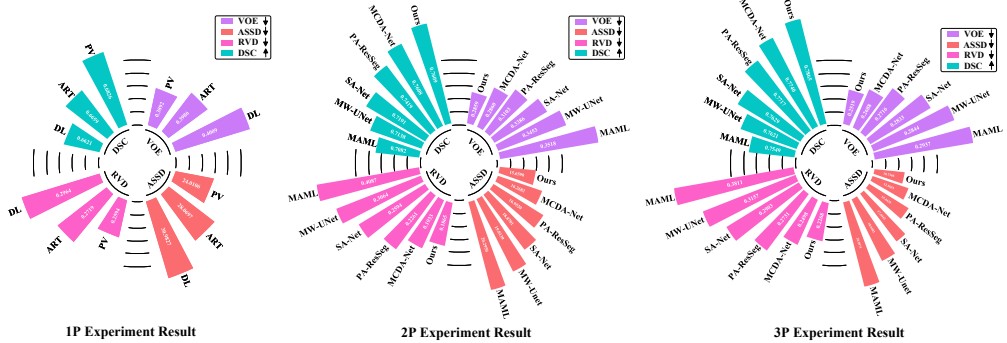

**1P Experiment Result**    **2P Experiment Result**    **3P Experiment Result**

Figure 4: Comparison of the segmentation performance on 1P, 2P and 3P, evaluated using multiple quantitative metrics including DSC, VOE, RVD, and ASSD. Specifically, the black upward arrow (↑) denotes that a higher metric value indicates superior performance, while the black downward arrow (↓) signifies that a lower metric value reflects more favorable outcomes.

Table 4: Quantitative comparison of 2-phase (left) and 3-phase (right) inputs. "Phase" indicates the data modality used. The best are highlighted in **bold**.

| Method | 2-Phase (A+P) | | | | 3-Phase (A+P+D) | | | |
|--------|---------|---------|---------|---------|---------|---------|---------|---------|
| | DSC(%)↑ | JSC(%)↑ | HD$_{95}$ (mm)↓ | ASSD (mm)↓ | DSC(%)↑ | JSC(%)↑ | HD$_{95}$ (mm)↓ | ASSD (mm)↓ |
| MAML | 70.82$_{↓5.27}$ | 64.82$_{↓6.59}$ | 53.223$_{↑24.38}$ | 20.295$_{↑4.64}$ | 75.49$_{↓3.16}$ | 70.63$_{↓4.18}$ | 41.588$_{↑14.78}$ | 20.097$_{↑9.36}$ |
| MW-UNet | 71.38$_{↓4.71}$ | 65.47$_{↓5.94}$ | 37.519$_{↑8.68}$ | 19.013$_{↑3.35}$ | 76.21$_{↓2.44}$ | 71.56$_{↓3.25}$ | 35.221$_{↑8.42}$ | 18.640$_{↑7.90}$ |
| SA-Net | 71.91$_{↓4.18}$ | 66.14$_{↓5.27}$ | 36.946$_{↑8.11}$ | 18.870$_{↑3.21}$ | 76.29$_{↓2.36}$ | 71.67$_{↓3.14}$ | 34.126$_{↑7.32}$ | 17.444$_{↑6.71}$ |
| PA-ResSeg | 74.19$_{↓1.90}$ | 68.97$_{↓2.44}$ | 31.287$_{↑2.45}$ | 16.952$_{↑1.29}$ | 77.17$_{↓1.48}$ | 72.84$_{↓1.97}$ | 33.608$_{↑6.80}$ | 15.443$_{↑4.71}$ |
| MCDA-Net | 76.08$_{↓0.01}$ | 71.40$_{↓0.01}$ | 30.436$_{↑1.60}$ | 16.268$_{↑0.61}$ | 77.40$_{↓1.25}$ | 73.12$_{↓1.69}$ | 28.600$_{↑1.79}$ | 12.565$_{↑1.83}$ |
| `MADF-Net` | **76.09** | **71.41** | **28.8382** | **15.6590** | **78.65** | **74.81** | **26.8068** | **10.7366** |

**Obs. ❹: 3P Fusion: ART, PV, and DL.** As listed in Table 4, the proposed `MADF-Net` achieved the state-of-the-art performance in terms of DSC (78.65%), JSC (74.81%), $HD_{95}$ (26.806), and ASSD (10.736) compared to the other five methods on the 3P fusion strategy. This indicates that our `MADF-Net` more accurately localizes the spatial positions and delineates the geometric shapes of the target regions. The quantitative comparison of the performance across 1P, 2P, and 3P fusion strategies is further illustrated in Figure 4, and the visualization of the segmentation results is shown in Figure 5. Compared to other methods, the proposed `MADF-Net` achieves the closest performance to the ground truth in terms of tumor contour localization and small-object boundary segmentation.

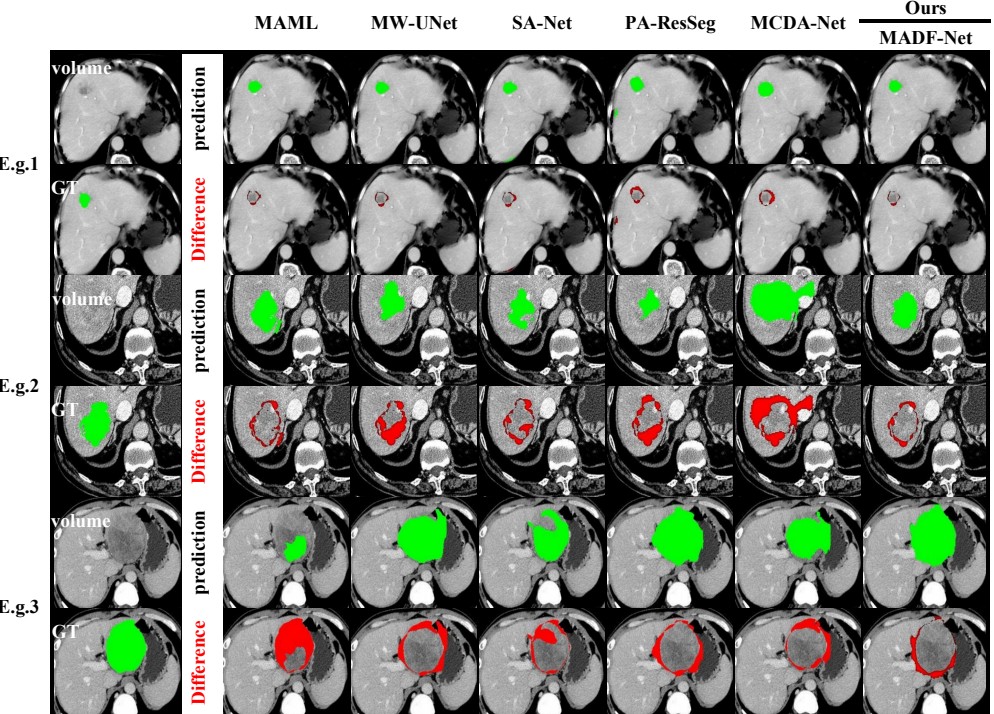

Figure 5: Result comparison of different three-phase networks. For better visualization, we performed appropriate cropping. The green region in the ground truth (GT) row represents the tumor, the green region in the prediction row indicates the predicted tumor area, and the red region in the difference row denotes the difference between the two.

## 5.2 EFFICIENCY ANALYSIS

Table 5 compares `MADF-Net` with baseline models. Our model requires $99.552 \times 10^9$ GFLOPs and 40.482 M parameters, achieving a favorable balance between computational cost and model size compared to SA-Net ($152.965 \times 10^9$ GFLOPs, 170.852 M) and PA-ResSeg ($64.660 \times 10^9$ GFLOPs, 67.732 M), while remaining competitive with lighter models such as MAML and MW-UNet. Further details on the experimental setup and efficiency comparisons are provided in Appendix C.

## 6 CONCLUSION AND FUTURE WORK

This paper presented `MADF-Net`, a novel multi-phase attention-based fusion network for liver tumor segmentation in contrast-enhanced CT images. Our approach is guided by a systematic quantitative evaluation of individual phases, confirming the predominant contribution of the PV phase and its alignment with clinical understanding. `MADF-Net` performs full-stage fusion across the input, feature, and decision levels to fully exploit the complementary information from ART, PV, and DL phases. The experiment on MPLL datasets demonstrate that our method achieves state-of-the-art performance and generalizes well across datasets. **Future Work:** We will (i) design phase-specific subnetworks and investigate phase-aware pretraining, and (ii) extend the paradigm to multi-modal/multi-omics fusion by integrating CT with digital pathology and radiomics-derived omics for comprehensive patient-level modeling.

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

## A ALGORITHM.

The overall workflow of the `MADF-Net` is summarized in algorithm 1.

---

**Algorithm 1:** High-level pseudocode of the proposed `MADF-Net`.

---

**Input:** Phase images: $I_A, I_B, I_C$; Clinical prior structure as a list of relations $R$; Reference **end**
mask $T$ (only for training)

**Output:** Segmentation output $Y_{\text{out}}$

1 Initialize network parameters $P$; set mode flag isTrain
2 **foreach** *batch in training/eval* **do**
   /* basic preprocessing and branch init                                              */
3   $I'_A \leftarrow \texttt{PrepData}(I_A)$
4   $I'_B \leftarrow \texttt{PrepData}(I_B)$
5   $I'_C \leftarrow \texttt{PrepData}(I_C)$
6   $I_m \leftarrow \texttt{CombineInit}(I'_A, I'_B, I'_C)$                     // early merged input for main branch
7   BranchInputs $\leftarrow$ [ $I'_A, I'_B, I'_C, I_m$ ]
   /* encode each branch to produce multi-level maps                                    */
8   **foreach** *entry J in BranchInputs* **do**
9     FeatureMaps[$J$] $\leftarrow \texttt{EncodeBlock}(J)$  // returns list of maps at depths 1..D
10  **end**
   /* clinical-aware message propagation (hierarchical depths)                          */
11  **for** *depth d = 1* **to** *D* **do**
12    **for** *node n in RelationOrder(R)* **do**
13      Parents $\leftarrow$ GetParents(n, $R$)
14      MsgIn $\leftarrow \texttt{PassMessages}$ ( [ FeatureMaps[parent][d] for parent in Parents ] )
15      UpdatedMap[n][d] $\leftarrow$ UpdateUnit( FeatureMaps[n][d], MsgIn )
          // residual-style update
16    **end**
17  **end**
   /* per-phase local refinement and prepare temporal stack                             */
18  **for** *phase p in* $\{A,B,C\}$ **do**
19    LowFeat $\leftarrow$ ExtractLow( UpdatedMap[p] )
20    Refined[p] $\leftarrow \texttt{LocalRefine}(LowFeat)$
21    StageOut[p] $\leftarrow$ ProjectForTemporal( Refined[p] )
22  **end**
23  TemporalStack $\leftarrow$ Stack( StageOut[A], StageOut[B], StageOut[C] )
   /* per-pixel temporal attention                                                      */
24  TemporalEnhanced $\leftarrow \texttt{TemporalPerPixel}(TemporalStack)$
   /* neighbor-aware cross-temporal fusion per phase                                    */
25  **for** *phase p in* $\{A,B,C\}$ **do**
26    Attended[p] $\leftarrow \texttt{NeighborInteract}$ ( UpdatedMap[p], TemporalEnhanced )
          // neighbor queries + relative-pos bias
27    Mix[p] $\leftarrow$ BlendLinear( Attended[p], $UpdatedMap[p]$) // weighted linear mixing
28    FinalFeat[p] $\leftarrow \texttt{ChannelBoost}$ ( Mix[p] )            // channel gating / enhancement
29  **end**
   /* merge multi-phase features and decode                                             */
30  MergedFeat $\leftarrow \texttt{FinalMerge}$ ( FinalFeat[A], FinalFeat[B], FinalFeat[C],
    UpdatedMap[$I_m$]) $Y_p red \leftarrow \texttt{DecodeBlock}$ ( MergedFeat )  // decoder with gated skips
   /* auxiliary outputs aggregation (if enabled)                                        */
31  AuxList $\leftarrow$ GetAuxOutputs()                              // possibly per-phase decoder heads
32  $Y_{\text{out}} \leftarrow$ OutputMerge( $Y_p red$, $AuxList$)
   /* loss and update (training only)                                                   */
33  **if** *isTrain* **then**
34    LossVal $\leftarrow \texttt{CalcLoss}$ ( $Y_{\text{out}}, T$ )
35    Backpropagate(LossVal, P)
36  **end**
37  **return** $Y_{\text{out}}$

---

Given the three single-phase CT images, the network first performs input-level preprocessing and branch initialization, generating both phase-specific and early-fused representations. These features are then processed through hierarchical encoder blocks, where clinical-prior message propagation is applied to capture inter-phase dependencies. Next, local refinement modules enhance low-level cues, followed by per-pixel temporal attention to model cross-phase temporal correlations. Neighbor-aware cross-temporal fusion and channel enhancement further integrate complementary information before multi-phase features are merged and decoded. Finally, decision-level aggregation combines auxiliary and main-branch outputs to produce the final segmentation mask. During training, the predicted mask is supervised by the ground-truth labels via a composite loss function.

## B  THE MPLL DATASET.

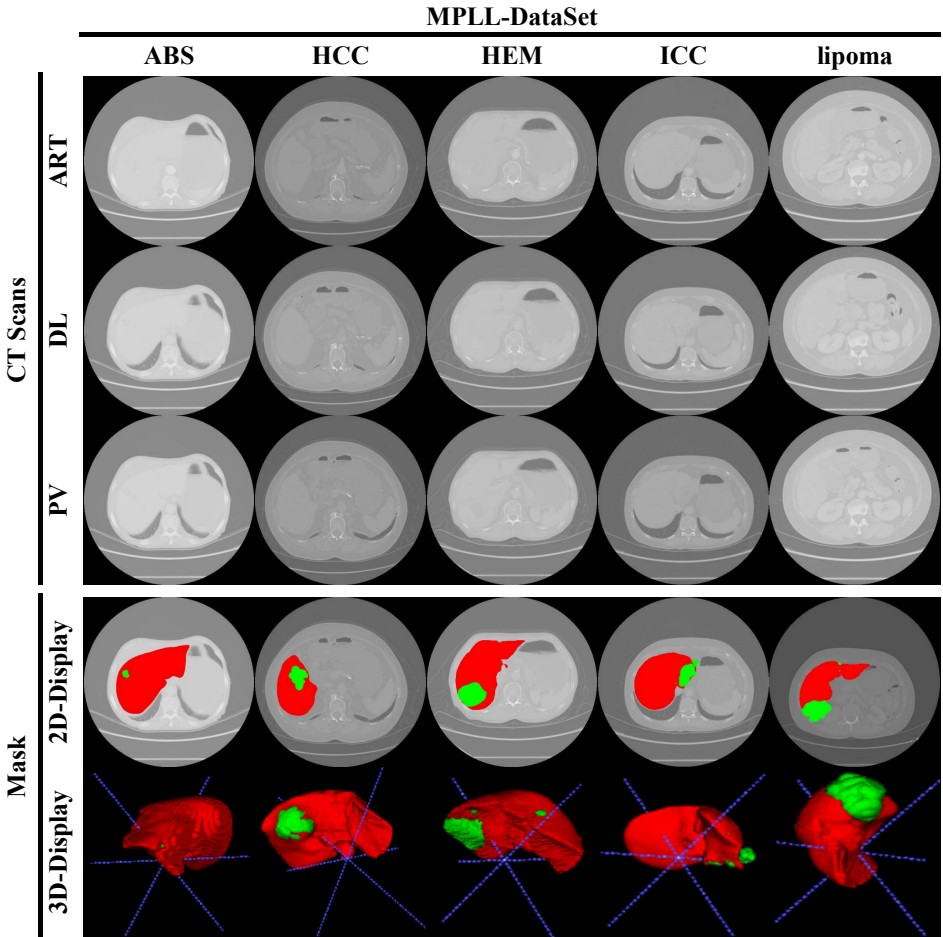

Figure 6: Example images from the MPLL dataset (red indicates liver regions, green indicates tumor regions).

❶: **Patient Cohort and Imaging Protocol.**  The Multi-Phase Liver Lesion (MPLL) dataset was collected at the "Anonymous Authoritative Hospitals (information will be made public after the paper is accepted)". The dataset comprises $952, 601$ 2D slices, making it one of the largest publicly reported multi-phase CT resources for liver tumor segmentation research, comprising 141 patients diagnosed with a wide spectrum of hepatic diseases. Imaging was performed between 2018 and 2022, covering both pediatric and adult populations (ages 9–72 years). All cases underwent standardized multi-phase contrast-enhanced CT examinations that included arterial, portal venous, and delayed phases, thereby capturing complementary hemodynamic information. Each scan was acquired at an in-plane resolution of $512 \times 512$ pixels, while slice thickness ranged from 0.62 mm to 5.0 mm. Due to differences in anatomical coverage, the number of slices varied considerably across

patients (48–777 slices per study). Data were de-identified before release, with ethical approval obtained in advance.

**❷: Clinical Diversity and Pathology Spectrum.** MPLL was intentionally designed to reflect real-world clinical heterogeneity. It contains patients diagnosed with common malignant tumors such as hepatocellular carcinoma (HCC) and intrahepatic cholangiocarcinoma, alongside a range of benign lesions including cysts, hemangiomas, and abscesses. This diverse pathology coverage ensures that the dataset does not disproportionately represent a single disease entity, but instead provides a representative benchmark for developing algorithms that are robust across varying lesion types, morphologies, and enhancement characteristics.

**❸: Dataset Organization and Splitting Strategy.** To support reproducible research, the dataset was partitioned into training, validation, and testing cohorts following a 7:1:2 split protocol consistent with contemporary studies (Jiang et al., 2023). Importantly, the test set was fixed to 30 cases and completely withheld during model design and training, thereby guaranteeing unbiased evaluation. This design facilitates fair performance comparison across different methods and helps prevent information leakage during algorithm development.

**❹: Preprocessing and Annotation Pipeline.** One critical challenge of multi-phase imaging is the misalignment across arterial, portal venous, and delayed acquisitions caused by respiration, patient movement, or cardiac activity. To mitigate this, a B-spline deformable registration strategy was applied using the portal venous phase as reference. This procedure significantly reduces inter-phase variability and enables spatially consistent feature fusion. Ground-truth lesion masks were annotated in ITK-SNAP by two board-certified radiologists, followed by an adjudication step by a senior radiologist. This three-stage process was designed to maximize accuracy, reduce annotation bias, and enhance inter-observer agreement.

**❺: Comparative Advantages over Existing Datasets.** Unlike widely used liver CT datasets such as LiTS2017 (Bilic et al., 2023) and Medical Segmentation Decathlon (Task 3: Liver), which primarily focus on single-phase CT, MPLL offers multi-phase contrast-enhanced imaging across arterial, portal venous, and delayed phases. This temporal richness provides unique opportunities for investigating cross-phase fusion strategies, which are critical for accurate lesion delineation but are underexplored in existing benchmarks. Moreover, MPLL is substantially larger in terms of slice count (over 950k slices), contains a broader age range including pediatric cases, and offers a more diverse pathology spectrum that includes both malignant and benign liver lesions. The dataset therefore not only complements but also surpasses existing resources in its ability to support the development of clinically relevant and generalizable liver lesion segmentation methods.

**❻: Dataset Significance.** In summary, MPLL represents a large-scale, carefully curated, and clinically diverse benchmark for multi-phase liver lesion segmentation. Its strengths lie in the combination of temporal imaging information, broad pathology spectrum, rigorous preprocessing, and high-quality expert annotations. Together, these characteristics make MPLL an invaluable resource for advancing multi-phase fusion strategies in medical image analysis. Representative examples highlighting inter-phase contrast variations and lesion depiction are illustrated in Figure 2.

Table 5: Efficiency Comparison of `MADF-Net` and Baseline Models (GFLOPs and Parameters). Performance is evaluated on the MPLL dataset under the 3-phase experiment setting. The bold indicates the best.

| Model | Gflops ($\times 10^9$) | Parameters (M) | Performance (%) |
|---|---|---|---|
| MAML (Zhang et al., 2021a) | **23.802** | 4.216 | 75.49 |
| MW-UNet (Zhu et al., 2022) | 53.419 | **2.773** | 76.21 |
| SA-Net (Zhang et al., 2021b) | 152.965 | 170.852 | 76.29 |
| PA-ResSeg (Xu et al., 2021) | 64.660 | 67.732 | 77.17 |
| MCDA-Net (Kuang et al., 2024) | 89.480 | 48.717 | 77.40 |
| Ours | 99.552 | 40.482 | **78.65** |

## C    DETAILED EFFICIENCY ANALYSIS.

As shown in Table 5, `MADF-Net` demonstrates a favorable balance between computational complexity, model capacity, and segmentation accuracy. While lightweight models such as MW-UNet achieve a small parameter size (2.773 M), they still require a non-trivial computational cost (53.419 GFLOPs) and their performance (76.21%) remains noticeably lower than ours. Similarly, MAML achieves the lowest GFLOPs (23.802) but suffers from a relatively limited accuracy (75.49%), which constrains its clinical applicability. On the other hand, heavier architectures like SA-Net and PA-ResSeg demand extremely large computational budgets (up to 152.965 GFLOPs and 170.852 M parameters), yet the corresponding accuracy (76.29% and 77.17%, respectively) provides only marginal improvement over lightweight baselines.

`MADF-Net` maintains a moderate parameter count of 40.482 M and a competitive computational demand of 99.552 GFLOPs, while delivering the **highest segmentation accuracy** (78.65%) among all compared methods. This clearly illustrates the *efficiency–accuracy trade-off*: although `MADF-Net` is not the most lightweight in terms of FLOPs or parameters, it achieves the best performance, outperforming both lightweight and heavyweight counterparts. This balance highlights `MADF-Net`'s practicality for real-world clinical deployment, where both computational feasibility and reliable accuracy are crucial.

## D    LARGE LANGUAGE MODELS USAGE STATEMENT

LLMs were used only for language polishing in this work. The manuscript was drafted entirely by the authors, and LLMs were employed solely to refine grammar and clarity of English expression. All scientific ideas, methods, and results are original contributions of the human authors, with LLM assistance limited to post-writing editing akin to traditional proofreading.

