# OpenReview forum: "Unlocking Clinical Potential: Beyond Single-to-Tri-Phase CT with Dynamic Fusion for Precision Liver Tumor Segmentation"
_ICLR.cc/2026/Conference — ICLR 2026 Conference Withdrawn Submission_

### Official Review · Reviewer_6vBo · 2025-10-19

**Soundness:** 2
**Presentation:** 2
**Contribution:** 2
**Rating:** 2
**Confidence:** 5

**Summary:**

This paper focuses on multi-phase liver tumor segmentation using contrast-enhanced CT (CECT) imaging. The authors first conduct a systematic quantitative evaluation of three imaging phases—arterial (ART), portal venous (PV), and delayed (DL)—to analyze their individual contributions to tumor segmentation. Based on these findings, they propose a Multi-phase Attention Deep Fusion Network (MADF-Net), which hierarchically integrates information from the three phases at the input, feature, and decision levels. In addition, the authors introduce a new large-scale, clinically diverse Multi-phase Liver Lesion (MPLL) dataset, which serves as the foundation for their experiments. Results demonstrate that MADF-Net achieves state-of-the-art performance, outperforming single-phase baselines and validating the effectiveness of deep multi-phase fusion for liver tumor segmentation.

**Strengths:**

1. The proposed dataset is significantly larger in scale compared with previous datasets. In summary, MPLL represents a large-scale, carefully curated, and clinically diverse benchmark for multi-phase liver lesion segmentation, contributing meaningfully to the advancement of this research area.
2. The supplementary materials provide detailed information about the dataset, including its collection process, annotation protocol, and data distribution across different phases. This comprehensive documentation enhances the transparency and reproducibility of the work, making the dataset a valuable resource for future studies.

**Weaknesses:**

1. The introduction of related work in both the Single-Phase–based and Multi-Phase–based Liver Tumor Segmentation sections is too limited. It only covers CNN-based methods and does not discuss recent popular frameworks such as Transformer, Mamba, or SAM. Moreover, the cited works are outdated and fail to reflect current progress in the field.
2. There are writing issues. One of the claimed contributions of this paper is the proposal of a new three-phase lung CT dataset; however, this contribution is not clearly stated in either the abstract or the introduction.
3. The proposed method lacks novelty. The three-stage fusion framework, whether at the input level, feature level, or decision level, shows little distinction from existing common fusion methods and does not demonstrate substantial innovation.
4. The experimental setup is insufficient. In Table 3, the compared methods on the single-phase dataset are too outdated, consisting mainly of simple early CNN models. The performance comparison with more recent and competitive methods is missing. Similarly, for the multi-phase fusion experiments, the baseline methods are also outdated, and no comparison is made with recent approaches from 2024 or 2025.
5. The ablation study is inadequate. It only compares the performance of using a single phase, two phases, and three phases. There is no detailed ablation analysis on the structure or components of the proposed three-stage fusion module itself.

**Questions:**

1. The writing is somewhat verbose. The three-stage fusion in the fusion task is a very common and conventional approach; presenting this part as Preliminary adds little to the paper and does not provide meaningful insight.
2. Figure 5 (visualization) should be revised. It is suggested to combine the difference and prediction results into a single figure—using green to indicate regions correctly segmented compared with the ground truth (GT), and red to indicate incorrectly segmented areas.
3. The entire paper should incorporate and verify results with more recent methods. Specifically, this includes updating: The related approaches discussed in the Introduction; The single-phase and multi-phase fusion methods reviewed in the Related Work section; The comparative methods used in the Experiments. Incorporating up-to-date techniques would significantly enhance the rigor and relevance of the work.

---

### Official Review · Reviewer_nCGX · 2025-10-23

**Soundness:** 2
**Presentation:** 2
**Contribution:** 2
**Rating:** 2
**Confidence:** 5

**Summary:**

This paper addresses the limitations of liver tumor segmentation from single-phase computed tomography. The authors leverage multi-phase contrast-enhanced CT, utilizing the arterial, portal venous, and delayed phases for complementary information. The study presented a quantitative analysis of each phase's individual contribution to segmentation, concluding that the PV phase is the most significant. The authors propose the **Multi-phase Attention Deep Fusion Network**, a model designed to hierarchically integrate features from all three phases across the input, feature, and decision levels. Experiments were conducted on a newly collected clinical dataset (MPLL). The results show that MADF-Net achieves a Dice score of 78.65%, and demonstrates consistent improvement across all evaluation metrics.

**Strengths:**

1. This paper is well-motivated, addressing clear limitations of single-phase CT for liver tumor segmentation. The rationale for using multi-phase contrast-enhanced CT is sound and directly addresses a significant clinical need.

2. The authors present a clear and easy-to-follow roadmap. They first conduct a systematic quantitative analysis of each CT phase, grounding the model's design in empirical evidence and clinical insights.

3. The proposed MADF-Net features a hierarchical fusion strategy that integrates information across input, feature, and decision levels.

4. The validation on the MPLL dataset is described as the largest and most comprehensive dataset for liver tumor segmentation. The authors also provided available code.

**Weaknesses:**

1. Several clinical and technical aspects remain unclear and flawed. The authors claim MPLL as the largest dataset with 952,601 2D slices, but it only comprises 141 patients. Multiple existing datasets contain significantly more patients (for example: https://vindr.ai/datasets/abdomen-phases, https://zenodo.org/records/12741586, and other research works [1,2,3]). Reporting the number of 2D slices is not a valid metric, as slice thickness varies, and the actual number of liver-containing slices would be considerably lower. Furthermore, the proposed 2D-based methodology is outdated, as current state-of-the-art liver tumor segmentation methods predominantly utilize 3D approaches. The comparison methods selected by the authors are also not current.

2. While the authors focus on AP, VP, and DP phases, they overlook the importance of the non-contrast phase in hepatic radiological diagnosis. The Hounsfield Unit enhancement between AP and non-contrast phases is a crucial diagnostic factor, and non-contrast imaging is essential for identifying calcifications.

3. The complexity of liver lesion types is not adequately addressed. Although the authors collected five lesion types (two malignant, three benign), their evaluation metrics lack clarity. They report only binary metrics combining all lesions as one type, without providing subtype analysis or malignant-benign differentiation. The evaluation focuses solely on segmentation performance, neglecting the clinically crucial aspect of lesion classification. Moreover, the dataset description lacks essential details about the distribution of patients with and without lesions, making the overall methodology unclear.

4. The insights proposed by the authors are well-established in existing literature and do not present novel perspectives. Based on the technical and clinical contribution, this paper is not suitable for ICLR.

[1] Ying, H., Liu, X., Zhang, M., Ren, Y., Zhen, S., Wang, X., ... & Cai, X. (2024). A multicenter clinical AI system study for detection and diagnosis of focal liver lesions. Nature Communications, 15(1), 1131.
[2] Wei, Y., Yang, M., Zhang, M., Gao, F., Zhang, N., Hu, F., ... & Liu, M. (2024). Focal liver lesion diagnosis with deep learning and multistage CT imaging. Nature communications, 15(1), 7040.
[3] Balaguer-Montero, M., Morales, A. M., Ligero, M., Zatse, C., Leiva, D., Atlagich, L. M., ... & Perez-Lopez, R. (2025). A CT-based deep learning-driven tool for automatic liver tumor detection and delineation in patients with cancer. Cell Reports Medicine, 6(4).

**Questions:**

1. How did the authors select AP, VP, and DP phases, considering that liver CE-CT typically includes both early and late arterial phases? The rationale for phase selection needs clarification.

2. The authors have not addressed the registration methodology. There is no discussion of how image registration accuracy was ensured across different phases.

3. What are the volume distributions and lesion subtypes in the dataset? This information is crucial, considering that prominent lesions such as HCC and hemangioma are often large in size, potentially making their segmentation less challenging.

---

### Official Review · Reviewer_CLa4 · 2025-10-28

**Soundness:** 2
**Presentation:** 2
**Contribution:** 2
**Rating:** 2
**Confidence:** 4

**Summary:**

The paper introduces MADF Net, a model developed for accurate liver tumor segmentation using multi phase contrast enhanced CT images, specifically the ART, PV, and DL phases. MADF Net integrates features hierarchically through attention mechanisms at three input, feature, and decision levels effectively addressing the limitations of conventional single stage fusion. Using the large and clinically comprehensive MPLL dataset, the proposed method achieves state of the art performance. This deep fusion strategy effectively utilizes complementary information from all three phases, resulting in more robust and precise lesion delineation.

**Strengths:**

The proposed MADF-Net performs deep feature fusion across three input, feature, and decision levels effectively overcoming the limitations of conventional single stage fusion methods. The study is supported by a systematic quantitative analysis that identifies the PV phase as the primary contributor to segmentation accuracy, ensuring that the model design aligns with clinical insights. The preliminary analysis is clearly presented and provides strong intuitive understanding. The method achieves superior segmentation performance on the test set. In addition, the paper introduces the MPLL dataset, a large and comprehensive CECT benchmark that enables robust evaluation and facilitates future multi phase research.

**Weaknesses:**

The paper presents several limitations regarding validation scope and methodological clarity. A major drawback is the lack of external validation, as the method is verified solely on the newly created MPLL dataset without rigorous testing against established public benchmarks. Methodologically, the final implementation often lacks a clear, one-to-one mapping or intuitive bridge to the mathematical rigor presented in the Preliminary section, particularly concerning the explicitly defined uncertainty-aware fusion and adaptive input weights, obscuring the precise realization of these concepts. Furthermore, the paper omits a comprehensive ablation study on all possible phase input combinations (e.g., comparing ART vs DL), hindering a precise understanding of the marginal value of each phase. Finally, the complex multi-stage attention network, despite its high accuracy, introduces a significant computational overhead (high GFLOPs), posing a practical limitation for clinical deployment.

**Questions:**

The validation of MADF-Net relies exclusively on the proprietary MPLL dataset. Although this dataset is comprehensive, the absence of external validation on well-established public benchmarks (e.g., LiTS, 3Dircadb) considerably limits the generalizability of the reported state of the art (SOTA) performance. Since ICLR is a general machine learning conference rather than one focused on medical imaging, broader validation across multiple datasets would strengthen the paper’s claims.

In addition, the study primarily investigates 1P (PV), 2P (ART+PV), and 3P (ART+PV+DL) settings. However, other important two-phase combinations (e.g., ART+DL or PV+DL) and the standalone contribution of the DL phase were not systematically evaluated. This omission limits a clear understanding of each phase’s marginal contribution to overall performance.

Furthermore, the general fusion formulations presented in the Preliminary section do not align directly with the implementation details described in the Methodology. While it is understandable that the implementation may involve additional technical complexity, each fusion module within the framework should be explicitly linked to the mathematical formulation introduced earlier. Otherwise, the inclusion of a complex adaptive formulation in the Preliminary section appears disconnected and underutilized in the final implementation, which causes confusion.

---

### Official Review · Reviewer_tPCv · 2025-11-01

**Soundness:** 3
**Presentation:** 3
**Contribution:** 2
**Rating:** 4
**Confidence:** 4

**Summary:**

This paper proposes a framework for liver segmentation, which integrates the information from three phases of CT images. Extensive experiments are conducted to demostrate the effectiveness of the method.

**Strengths:**

1. The presentation is clear
2. The idea is reasonable and achieves good performance according quality and quantity results

**Weaknesses:**

1. The scope of the method is too narrow; it only focuses on liver tumor segmentation. I think it is better to see how the method can perform in other medical tasks with CT images, like brain disease or lung disease segmentation

2. The method involves more parameters and data for better results, which I think may cost more computing time than other methods.

3. Although MADF-Net achieves state-of-the-art performance on the MPLL dataset, all experiments were conducted exclusively on this internal, non-public dataset collected from anonymous hospitals. The absence of external validation on publicly available benchmarks such as LiTS or MSD limits the generalizability of the conclusions.

**Questions:**

Why do the two-phase studies only consider ART+PV? What about other combinations?

---

### Official Review · Reviewer_zjKY · 2025-11-03

**Soundness:** 2
**Presentation:** 2
**Contribution:** 1
**Rating:** 2
**Confidence:** 4

**Summary:**

The paper introduces a Multi-phase Attention Deep Fusion Network (MADF-Net) designed for liver tumor segmentation using contrast-enhanced CT (CECT) data. This CECT data includes three enhancement phases, including arterial (ART), portal venous (PV), and delayed (DL). At the input level, the network performs channel-level concatenation of the three phases. For feature-level fusion, it employs a gated residual summation mechanism, while decision-level fusion is achieved through convolution-based aggregation. The authors also build a new dataset (MPLL) with three-phase CECT images and report a Dice of 78.65%, improving by 9.39% over single-phase baselines.

**Strengths:**

1. The proposed MPLL dataset is relatively comprehensive and clinically diverse, covering 141 patients and multiple tumor types.
2. The rationale for using multi-phase data (ART, PV, DL) is well-grounded in radiological knowledge and justified by quantitative phase-wise evaluation.

**Weaknesses:**

1. Lack of true methodological novelty. The proposed MADF-Net mainly combines input-level concatenation, attention-based feature fusion, and decision-level aggregation. These strategies have been well explored in prior works such as PA-ResSeg (Xu et al., 2021), SA-Net (Zhang et al., 2021b), and MCDA-Net (Kuang et al., 2024). While the integration of these three fusion stages is labeled as hierarchical, the design essentially stacks existing fusion strategies rather than introducing a conceptually new mechanism. Overall, the novelty is incremental and would be considered engineering-level improvement.

2. Unclear contribution of each fusion stage. Although the paper emphasizes multi-level fusion, it lacks an ablation study isolating the contribution of input-, feature-, and decision-level fusion components. Without such evidence, it is hard to assess whether the performance gain stems from the combination itself or simply from increased model capacity.

3. Dataset not publicly validated. The proposed MPLL dataset is claimed to be the largest multi-phase liver dataset, yet the paper provides no cross-center validation or public access. The clinical value of the dataset would be more effectively demonstrated by comparing it with existing public datasets and testing the generalizability of the method trained on this dataset using an external cohort.

4. Limited insight into dynamic fusion. Despite the title suggesting Dynamic Fusion, the proposed attention modules are static softmax-based weights and do not dynamically adapt to phase availability or uncertainty at inference time. In clinical deployment, missing or misaligned phases are common, so robustness to incomplete-phase input should be also demonstrated.

**Questions:**

1. Will the accuracy of registration between different phase images impact the fusion process? How to ensure that the registration is sufficiently precise?
2. During the annotation process, how do the attending radiologists annotate liver tumors? Which phase images do they utilize for this task, and how do they combine these images? Furthermore, how is annotation inconsistency among the three radiologists resolved? What strategies are employed to address inconsistencies across different phase images?
3. What is the main difference between the proposed method and previous multi-stage fusion methods?
4. Why not fuse NCCT images with three-phase CECT images?
5. How do the experiments support the claim that the results "generalize well across datasets" in the conclusion?
6. Will the MPLL dataset be made available to the public?

---

### Note · Authors · 2025-11-12

I have read and agree with the venue's withdrawal policy on behalf of myself and my co-authors.